# Transactions between Problem Behaviors and Academic Performance in Early Childhood

**DOI:** 10.3390/ijerph19159583

**Published:** 2022-08-04

**Authors:** Chin-Chih Chen, Sheng-Lun Cheng, Yaoying Xu, Kathleen Rudasill, Reed Senter, Fa Zhang, Melissa Washington-Nortey, Nikki Adams

**Affiliations:** 1Department of Counseling and Special Education, Virginia Commonwealth University, Richmond, VA 23284, USA; 2Department of Library Science and Technology, Sam Houston State University, Huntsville, TX 77341, USA; 3School of Education, Virginia Commonwealth University, Richmond, VA 23284, USA; 4Department of Education, Open University of China, Beijing 100039, China; 5Department of Psychology, King’s College, London WC2R 2LS, UK; 6Ph.D. Program in Special Education, University of Illinois at Urbana-Champaign, Champaign, IL 61820, USA

**Keywords:** problem behaviors, academic performance, transactional relationship, early childhood

## Abstract

This study aimed to further the understanding of transactional relationships that exist between problem behaviors and academic performance in early childhood. Early academic and behavior difficulties increase the risk of school disengagement, academic failure, and dropout. Although children’s academic and behavioral difficulties have been shown to be intercorrelated, little research has focused on how the relationship reciprocates and progresses in early childhood. This study investigated how problem behaviors (i.e., externalizing and internalizing) influence and are influenced by academic performance (i.e., poor reading and math) from kindergarten to third grade. Participants included 18,135 students (51.22% boys) derived from a nationally representative sample in the Early Childhood Longitudinal Study, Kindergarten Class of 2011 (ECLS-K: 2011). Teacher ratings of children’s internalizing (low self-esteem, anxiety, loneliness, or sadness) and externalizing (fighting, arguing, showing anger, impulsively acting, and disruptive behaviors) problem behaviors, as well as direct assessments of children’s academic performance (reading and math), were collected yearly. Cross-lagged panel modeling (CLPM) was employed to examine reciprocal relationships between problem behaviors and academic performance over time from kindergarten to third grade. The results supported the transactional relationships in early childhood, with higher externalizing as well as internalizing problem behaviors predicting lower academic performance and lower academic performance predicting higher externalizing and internalizing problem behaviors. The implications for research, prevention, and early intervention regarding the progression of academic and behavioral problems are discussed.

## 1. Introduction

Early academic and behavioral struggles place children at a heightened risk for school disengagement, academic failure, and dropout, and they can be expected to lead to challenges during adulthood [1,2,3,4]. Emerging research suggests that behavioral and academic problems do not function unidirectionally but instead work reciprocally to predict later school maladjustment and failure [5]. However, findings on the reciprocal associations between problem behaviors and academic achievement for children are less conclusive. A nuanced understanding of the complex mechanisms between behavioral and academic problems could strengthen our capacity to assess and identify students’ barriers to learning and better inform the development of evidence-based intervention efforts.

Children’s problem behaviors and academic challenges likely exacerbate one another and place children at greater risk for maladaptive adjustment, with serious consequences for society [6,7]. Research has demonstrated a negative association between problem behaviors and academic problems [8,9,10]. Academic difficulties are predictive of behavior difficulties, including internalization (e.g., depression) and externalization (e.g., disruptive behavior). Similarly, students with internalizing and externalizing problem behaviors are at increased risk for lower academic performance (e.g., [11,12,13]). While some studies have clearly established unidirectional relationships between academic and behavioral problems, other research has postulated transactional relationships, which suggest that behavior and academic difficulties work reciprocally to predict later school maladjustment and failure [5,11].

However, there is a paucity of research examining the developmental progression of behavioral and academic problems and how these problems potentially reciprocate during early childhood, the period of a child’s most rapid growth/development. It is important to clarify the developmental processes underlying behavior (e.g., externalizing and internalizing) problems and academic performance for those facing early adversity and identify potential points of intervention to mitigate maladaptive adjustment over time. The proposed study investigates how internalizing and externalizing problem behaviors influence and are influenced by academic performance and vice versa from kindergarten to third grade in the hope of elucidating these transactional relationships in early childhood.

### 1.1. Transactions between Behavioral and Academic Problems

According to developmental and social–ecological theory, variations in individual functioning can trigger a sequence of events [14,15]. According to dynamic systems theory, human development involves a cascade of changes influenced by transactions within individuals or between individuals and their social contexts [16]. Poor behavior could lead to lower levels of academic functioning, and, in turn, academic failure could lead to vulnerability to future behavior issues. The link between the behavior and academic domains suggests the possibility of developmental cascades [5,11].

Several models have been proposed to explain the relationships between problem behaviors and academic performance, including the adjustment erosion model, academic incompetence model, bidirectional model, and shared risk model [11]. The adjustment erosion model posits that the behavioral construct at an earlier time point has a significant negative correlation with academic success at a later time point, while the academic incompetence model postulates that poor academic competence and performance predict later externalizing and internalizing problems. To demonstrate bidirectional associations between behavior and academic achievement, the behavioral construct must predict academic achievement at a later time point and vice versa. The shared risk model postulates an alternative hypothesis, which states that behavioral problems and academic struggles share a common predictive factor.

### 1.2. Externalizing Problem Behaviors and Academic Achievement

Considerable stability of problem behaviors has been identified beginning in early childhood. Externalizing problems, such as aggressive/disruptive behavior, can interfere with students’ classroom learning and affect classroom dynamics. Such linkage is evident even before school begins [17]. Recent longitudinal studies have aimed to identify transactions between academic achievement and externalizing problems and have revealed underlying processes across multiple developmental periods. With the use of cross-lagged modeling, these longitudinal studies have tested such transactional relationships by controlling for interindividual stability over time and within-time covariation of externalizing problems and academic performance. However, the findings on transactional effects are inconclusive.

Few studies have shown bidirectional relationships that suggest that externalizing behaviors have a negative effect on academic performance and vice versa from late childhood to adolescence. Moilanen and colleagues [11] studied cross-lagged effects of at-risk U.S. boys and found that higher externalizing problems in sixth and eighth grade were indicative of lower academic competence in eighth and tenth grade. Lower academic competence in 10th and 11th grade was predicative of higher externalizing problems in 11th and 12th grade. Zhang and colleagues [13] identified bidirectional associations between externalizing behaviors and academic performance (final exam scores) across fifth to ninth grade for Chinese children and noted that the predictive strength from academic failure to externalizing behaviors increased, whereas the magnitude of the adjustment erosion pathway decreased during the transition into adolescence. Zimmermann and colleagues [18] identified similar bidirectional relationships between externalizing problems and academic achievement (grades and standardized assessments) of German children from fifth to ninth grade. Externalizing problems had a more notable impact on grades than they had on standard achievement. Similarly, Okano et al. [19] found bidirectional effects between externalizing behaviors and school achievement from fifth to ninth grade for U.S. children. A longitudinal study of Canadian children found negative cross-lagged effects between externalizing behaviors and academic competence from age 8 to 9 to age 14 to 15 [20].

Studies also demonstrated bidirectional relationships from early childhood to mid-childhood. Spanning kindergarten to fourth grade, Metsäpelto and colleagues [21] identified bidirectional associations suggesting that high externalizing problems in earlier grades were linked to low academic performance in later grades and vice versa for Finnish children. Van der Ende and colleagues [22] examined both internalizing and externalizing behaviors and academic difficulties in a multiple-cohort longitudinal study spanning eight years of children in the Netherlands aged six and ten. The results indicated that externalizing behaviors predicted academic difficulties, and academic difficulties predicted both externalizing and internalizing problem behaviors.

Different from prior studies that have reported bidirectional pathways, many studies have only identified unidirectional relationships between externalizing problem behaviors and academic achievement. For example, several studies have strongly supported the adjustment erosion model, suggesting that externalizing behaviors predict academic difficulties from childhood through adolescence and into adulthood [5,23,24,25,26,27]. By contrast, Vaillancourt et al.’s [28] findings supported the academic incompetence model, suggesting that aggression predicted academic competence/performance during early/middle childhood (standardized assessment and GPA).

### 1.3. Internalizing Problem Behaviors and Academic Achievement

Findings on the transactions between internalizing symptoms and academic achievement are also mixed in the following studies, which used cross-lagged modeling. Some studies identified bidirectional linkages between depression, anxiety, and general internalizing symptoms and academic achievement over time. Keles and colleagues [29] identified reciprocal relations between general internalizing symptoms and academic ratings in a longitudinal cohort of Norwegian students from fourth to seventh grade. Verboom and colleagues [30] examined the associations between depressive symptoms and academic performance in a longitudinal study of Dutch adolescents between the ages of 10 and 18 years. However, they only found bidirectional associations in girls. Zhang and colleagues [13] found bidirectional associations between depression and academic achievement among Chinese students from grades five to nine, but the results were only observed in the narrow transition into middle school. Weidman and colleagues [31] also found bidirectional relationships between depression and anxiety and academic achievement (GPA) among U.S. children between sixth and tenth grade. Okano and colleagues [12] found significant bidirectional pathways between internalizing behaviors and academic grades among girls from third to ninth grade.

Most other longitudinal studies have identified unidirectional pathways between internalizing behaviors and academic achievement. Some studies have only supported adjustment erosion pathways, suggesting that internalizing behaviors predict academic achievement. For example, Hall and colleagues [32] found social withdrawal in kindergarten and first grade predicted reading achievement in second grade. There are many studies that only support the academic incompetence model and strongly suggest that academic achievement predicts later internalizing behaviors [5,11,22,24,26,33]; such associations were found in later years through adolescence [11,24,26], and into adulthood [22]. The pathway appears stronger in girls also [26].

### 1.4. Current Study

Early childhood is a time full of developmental opportunities and vulnerability. While the transactions between problem behaviors and academic performance have been documented [5,11,34], there is a paucity of longitudinal research examining the transactional associations in early childhood from a developmental perspective. Mixed findings exist in the current literature. Some studies have shown unidirectional links and others have established bidirectional links between externalizing and internalizing problem behaviors and academic performance across different developmental stages [5,11,18,22,25,30]. However, little research has specifically addressed the transactions between behavior problems and academic performance in early childhood.

For interventions to be successful in preventing the progression of academic and behavior problems, it is necessary to understand transactional relationships between behavior and academic issues over time [5,8,9,18,22]. This study investigated how internalizing and externalizing problems predict and are predicted by academic difficulties (i.e., poor academic achievement and classroom problem behavior) over time. We addressed the following research questions to unpack the developmental mechanism for students in early childhood: (1) Do externalizing problem behaviors predict children’s academic performance in K–3 over time and vice versa? (2) Do internalizing problem behaviors predict children’s academic performance in K–3 over time and vice versa? We hypothesized that there are transactional relationships between externalizing and internalizing problem behavior and academic performance.

## 2. Method

### 2.1. Participants and Research Design

This study consisted of 18,135 children derived from a nationally representative sample in the United States participating in the Early Childhood Longitudinal Study, Kindergarten Class of 2011 (ECLS-K: 2011) [35]. About 51.22% of the children were male (*n* = 9288) and 48.78% were female (*n* = 8847). Most of the children were White (46.81%; *n* = 8488), and most had two biological/adoptive parents (69.03%; *n* = 9249) with a high school diploma (22.14%; *n* = 3543) or some college education (26.50%; *n* = 4242). More than half of the children (52.27%; *n* = 7070) lived in a household at or above 200 percent of the poverty threshold. The average age of children was 73.44 months (SD = 4.47) (note: percentages and numbers listed above were unweighted).

On the basis of a multistage probability sample design, the ECLS-K: 2011 drew a nationally representative sample of U.S. children enrolled in kindergarten during the 2010–2011 school year and longitudinally followed the same sample through fifth grade (201–2016) [35]. The multistage probability sample design involved a three-stage process, starting with dividing the U.S. and sampling primary sample units (PSUs), then selecting both public and private schools within each PSU, and finally sampling a fixed number of kindergarten children within each sampled school. The complete dataset included 970 schools within 90 PSUs [35]. It should be noted that because of the design of the ECLS-K: 2011, teachers were included in the study by virtue of their connection to the sampled children. Therefore, there were no specific procedures implemented to recruit teachers.

Employing multiple sources and methods, the ECLS-K: 2011 provides researchers and policymakers with information on children’s early education experiences, development, and learning over time collected from parent interviews, teacher ratings/surveys, school records, and direct child assessments. In the current study, we used publicly available data collected each spring semester during kindergarten and the first-, second-, and third-grade years to analyze the longitudinal relationships between children’s problem behaviors and academic performance. We used direct child assessments of reading and mathematics and teacher questionnaires of children’s internalizing and externalizing problem behaviors.

### 2.2. Measures

Internalizing and Externalizing Problem Behaviors. Teachers were asked to rate how frequently children exhibited problem behaviors using two subscales from the modified version of the Social Skills Rating System (SSRS) [36] in the ECLS-K: 2011 study. Children’s behaviors were rated on a 4-point Likert scale, ranging from never to very often. Teachers rated 4 items related to the child’s internalizing behaviors (low self-esteem, anxiety, loneliness, and sadness) and 5 items for externalizing behaviors (fighting, arguing, showing anger, impulsively acting, and disrupting class activities). Average scores were created for each of the scales of internalizing and externalizing problem behaviors. The higher the scores, the more likely the child was to exhibit internalizing and externalizing problem behaviors. The Cronbach’s alpha coefficients for internalizing and externalizing problem behaviors were 0.89 and 0.78.

Academic Performance. Academic performance assessments that focused on children’s reading and mathematics skills were administered using a two-stage process. The ECLS-K used Item Response Theory (IRT) scale scores for the children’s assessments in reading and mathematics [35]. IRT scores were determined by the number of correct answers and the probability of the child answering the questions that were not administered correctly. Using the IRT, children’s scores can be compared within and across grades. The reading assessment measured children’s basic language skills, vocabulary knowledge, and reading comprehension. In the two-stage process, reading was first assessed with the measure of English basic reading skills (EBRS), which includes 18 reading items and 2 screener items from the Preschool Language Assessment Scale [37]. The scores on the EBRS determined the difficulty of the questions (low, middle, or high) that the child would receive in the second stage. Children who spoke Spanish and did not meet the criteria for the EBRS were evaluated by the measure of Spanish early reading skills (SERS), which encompassed 31 items translated from the low and middle second-stage English reading test.

The mathematics assessment measured children’s conceptual knowledge, procedural knowledge, and problem-solving. Items consisted of questions on number sense, properties, and operations; measurement; geometry and spatial sense; data analysis, statistics, and probability; and patterns, algebra, and functions. The first stage of the assessment included 18 items that were used to determine the difficulty of the questions (low, middle, or high) in the second-stage assessment [35]. Children whose home language was Spanish and who did not pass the preLAS were administered the entire mathematics assessment in Spanish. According to the user’s manual, the Cronbach’s alpha coefficients for reading and mathematics assessments were 0.95 and 0.94.

Demographics. We controlled for gender and ethnicity as covariates in this study. Gender was binary, with males as the reference group. Races/ethnicities were categorized as non-Hispanic White, black/African American, Hispanic, Asian, and “other”, with the non-Hispanic White group as the reference group.

### 2.3. Data Analysis

Prior to the main analyses, descriptive statistics and bivariate correlations were examined. Cross-lagged panel modeling (CLPM) was then employed to examine bidirectional relationships between problem behaviors and academic achievement across four timepoints—kindergarten to third grade—while controlling for gender and race/ethnicity. One of the primary goals of CLPM is to identify causal predominance by assessing the magnitude of the lagged effects while accounting for contemporaneous effects (i.e., autoregressive effects) and variance across time [38]. We built two cross-lagged panel models (CLPM) to examine the reciprocal relationships between internalizing/externalizing problem behaviors and academic achievement. Gender and race/ethnicity were included in all the CLPMs as covariates. In addition to the cross-lagged and autoregressive paths between adjacent time points, second-order autoregressive paths (AR2, see [39]) between the first and third time point measurements and between the second and fourth time point measurements were specified. These additional autoregressive pathways are reasonable for CLPMs with at least three measurement points because influences observed in the early stage may be carried over beyond the steady change between adjacent measurements [40,41]. The overall goodness of model fit was assessed by global fit statistics, including the comparative fit index (CFI), the Tucker–Lewis index (TLI), the root mean square error of approximation (RMSEA), and the standardized root mean square residual (SRMR). A satisfactory model fit was defined by the following criteria: CFI ≥ 0.95, TLI ≥ 0.95, RMSEA ≤ 0.06, and SRMR ≤ 0.08 [42].

Full information maximum likelihood (FIML) implemented in Mplus 8.4 [43] was used to handle missing data in endogenous variables because it is recommended as an ideal approach to address missingness in longitudinal panel models [44,45]. The TYPE = COMPLEX option in Mplus 8.4 was used to account for the complex sampling design employed by the ECLS-K: 2011. To obtain correct point estimates and produce accurate variance estimation, the child-level sampling weight (W7C27P_2T270) and 80 corresponding replicate weights (W7C27P_2T271-W7C27P_2T2780) were entered into the analytic models. The paired jackknife replication method (JK2) was selected, as recommended by the user’s manual for the ECLS-K: 2011 [35].

## 3. Results

### 3.1. Descriptive Statistics and Correlations

Table 1 presents descriptive statistics for problem behaviors and academic performance variables. The absolute values of skewness and kurtosis were all less than 1.5, suggesting that the assumption of normality was met [46] (Kline, 2016). Table 2 presents zero-order correlations. The directions of the coefficients aligned with the theory. Children’s earlier internalizing problem behaviors were strongly related to later internalizing problem behaviors (*r* ≥ 0.26, *p* < 0.001). Earlier externalizing problem behaviors were strongly related to later externalizing problem behaviors (*r* ≥ 0.53, *p* < 0.001). Similarly, earlier academic achievement was strongly related to later achievement (*r* ≥ 0.75). Internalizing and externalizing problem behaviors had positive associations with one another, with *r*s ranging from 0.08 to 0.33 (*p* < 0.001). Academic achievement had negative associations with internalizing and externalizing problem behaviors, with *r*s ranging from 0.14 to 0.23 (*p* < 0.001).

The first CLPM examined the longitudinal relationships between internalizing problem behaviors and academic achievement while controlling for gender and race/ethnicity. The model fit the data well ($\chi_{\left(18\right)}^{2}$ = 679.739, *p* < 0.001; CFI = 0.980, TLI = 0.937, RMSEA = 0.046, SRMR = 0.057). As shown in Figure 1, all the autoregressive paths were statistically significant. The autoregressive coefficients of internalizing problem behaviors between adjacent time points were stable over time (*β* = 0.273, K to first grade; *β* = 0.212, first to second grade; *β* = 0.262, second to third grade; *p* < 0.001). The autoregressive coefficients of academic achievement between adjacent time points were particularly large, which showed substantial stability over time (*β* = 0.833, K to first grade; *β* = 0.785, first to second grade; *β* = 0.733, second to third grade; *p* < 0.001). In addition, all the second-order autoregressive (AR2) pathways were also statistically significant. A consistent AR2 emerged for internalizing problem behaviors (*β* = 0.161, K to second grade; *β* = 0.185, first to third grade; *p* < 0.001) and academic achievement (*β* = 0.119, K to second grade; *β* = 0.179, first to third grade; *p* < 0.001). 

In terms of the cross-lagged relationships, internalizing problem behaviors and academic achievement consistently predicted one another over time, even after controlling for autoregressive paths, AR2, gender, and race/ethnicity. Internalizing problem behaviors negatively predicted academic achievement in reading and math from kindergarten to first grade (*β* = −0.052, *p* < 0.001), from first grade to second grade (*β* = −0.014, *p* < 0.001), and from second grade to third grade (*β* = −0.013, *p* < 0.001). The magnitude of the transactional effects of internalizing problem behaviors on academic achievement appeared significant but slightly weakened over time. Academic achievement also negatively predicted internalizing problem behaviors from kindergarten to first grade (*β* = −0.134, *p* < 0.001), from first grade to second grade (*β* = −0.170, *p* < 0.001), and from second grade to third grade (*β* = −0.113, *p* < 0.001), with the largest path from first grade to second grade.

### 3.2. Transactional Relationships between Externalizing Problem Behaviors and Academic Performance

The second CLPM examined the transactional relationships between externalizing problem behaviors and academic achievement longitudinally while controlling for gender and race/ethnicity. The model also fit the data well ($\chi_{\left(18\right)}^{2}$ = 1106.618, *p* < 0.001; CFI = 0.973, TLI = 0.913, RMSEA = 0.059, SRMR = 0.068). As shown in Figure 2, all the autoregressive paths were statistically significant. The autoregressive coefficients of externalizing problem behaviors between adjacent time points were stable over time (*β* = 0.567, K to first grade; *β* = 0.399, first to second grade; *β* = 0.393, second to third grade; *p* < 0.001). The autoregressive coefficients of academic achievement between adjacent time points were, as with the internalizing model, particularly large (*β* = 0.833, K to first grade; *β* = 0.786, first to second grade; *β* = 0.732, second to third grade; *p* < 0.001). In addition, all the second-order autoregressive pathways were also statistically significant. A consistent AR2 emerged for externalizing problem behavior (*β* = 0.294, K to second grade; *β* = 0.309, first to third grade; *p* < 0.001) and academic achievement (*β* = 0.118, K to second grade; *β* = 0.181, first to third grade; *p* < 0.001). 

In terms of the cross-lagged relationships, there were transactional effects between externalizing problem behaviors and academic performance over time. Externalizing problem behaviors significantly predicted academic achievement. Higher externalizing problem behaviors predicted lower academic achievement from kindergarten to first grade (*β* = −0.053, *p* < 0.001) and second to third grade (*β* = −0.015, *p* < 0.001) but not first to second grade. Academic achievement predicted externalizing problem behaviors at all time points. Specifically, greater academic achievement consistently predicted lower externalizing problem behaviors (*β* = −0.070, K to first grade; *β* = −0.045, first to second grade; *β* = −0.028, second to third grade; *p* < 0.001).

## 4. Discussion

Children’s academic and behavioral problems have been shown to be intercorrelated and reciprocally affected. This study is one of the first to investigate the transactional relationships between externalizing and internalizing problem behaviors and academic achievement in early childhood using a nationally representative sample. Using cross-lagged modeling and controlling for within-individual stabilities over time and covariance, we identified K to third grade as the time when developmental cascading occurs in the transactional process. Our discussion focuses on the understanding of the developmental mechanisms in terms of timing and transactional process and on leveraging this information in early intervention efforts to facilitate positive development.

### 4.1. Transactions in Early Childhood

Prior research has suggested that externalizing and internalizing problem behaviors and poor academic performance function reciprocally and contribute to later maladjustment and school failure (e.g., [11,13]). This study further identified bidirectional relationships between externalizing and internalizing problem behaviors and academic achievement in early childhood; these relationships were identified earlier than those observed in most prior longitudinal studies, which have identified bidirectional relationships between academic performance and externalizing problem behaviors (e.g., [11,13,18,19]) and internalizing problem behaviors (e.g., [13,29,30,31]) from middle childhood through adolescence.

Additionally, our findings are consistent with and build on findings from two kindergarten cohort studies [21,22], which showed transactional relationships between externalizing problem behaviors and academic achievement from kindergarten to third grade, as well as prior longitudinal studies, which also indicated bidirectional relationships between internalizing problem behaviors (e.g., [13,29,30,31]) and academic performance from middle childhood through adolescence. What is novel in the current study is the confirmation of these bidirectional relationships with a nationally representative sample of children. 

This study contributes to the existing literature on transactional relationships between problem behaviors and academic performance by identifying such relationships in early childhood. The findings highlight the importance of considering nuanced information in intervention efforts to manage students’ academic and behavior problems as early as kindergarten. Further attention should be paid to addressing the transactional nature of internalizing problem behaviors and academic performance.

### 4.2. Adjustment Erosion, Academic Incompetence, or Bidirectional

While most studies support adjustment erosion as the primary model in the investigation of externalizing problems and academic performance (e.g., Chen et al., 2010; Deighton et al., 2018; Zhou et al., 2010), this study provides evidence of a bidirectional, reciprocal linkage, suggesting that externalizing problem behaviors are related to change in academic failures and vice versa. As noted in prior research, few studies have found bidirectional pathways between externalizing behaviors and academic performance (e.g., [11,13,18,19,47,48]). Additional studies are needed that further explore the link between externalizing problem behavior and academic achievement in early childhood. Notably, and contrary to our findings from kindergarten to first grade and second grade to third grade, externalizing problem behaviors in first grade did not predict academic achievement in second grade. Future studies can also examine whether other factors covary with externalizing problem behaviors and affect academic achievement.

Research on the associations between academic performance and internalizing problem behaviors is less conclusive. The academic incompetence model, which posits that poor academic competence predicts problem externalizing and internalizing behaviors, has been validated from middle childhood to late adolescence by many researchers. Consistent with prior studies (e.g., [29,30,31]), the linkage between internalizing problem behaviors and academic performance has also been found to be bidirectional.

### 4.3. Implications in Research and Practice

This study delineated the timing and processes between problem behaviors and academic performance, which has important implications for guiding the targets and timing of intervention efforts. A nuanced understanding of the complex mechanisms between behavioral and academic problems could strengthen our capacity to assess and identify students’ barriers to learning and help inform the development of evidence-based intervention efforts. The investigation of transaction relationships is crucial not only for basic research but also for prevention/intervention efforts. It will help practitioners target domains of adjustment that will likely contribute to problems in other domains. Armed with the knowledge that these problem behaviors are more likely to interact with academic achievement, educators may be alert to the presentation of externalizing/internalizing symptoms and provide the support needed to those who are struggling with behavioral challenges and prevent more pervasive difficulties in other domains.

Educators should be mindful of these predictive associations and should strive to provide holistic support for students’ externalizing behaviors and internalizing symptoms alongside their academic challenges. Intervening across domains may forestall the “spread” of children’s difficulties; for instance, if a child presents with externalizing behavior problems, intervention in both behavior and academic performance may mitigate the risk of later academic problems. Due to the complex and significant associations between externalizing behaviors, internalizing symptoms, and academic challenges, educators may benefit from the implementation of a robust social-emotional curriculum as a complement to their academic instruction and interventions. 

### 4.4. Limitations and Further Research

Future research can provide further clarity to the directionality between academic performance and behavior with the consideration of the investigation of variables that moderate or mediate these associations. It is important to explore the salient student characteristics that may moderate the association between problem behaviors and academic achievement. This notion ties back to the shared risk theory, which presents the possibility that the association between academic performance and behavior may be explained by another variable (e.g., gender, age, race, socioeconomic status, or family characteristics). Future research can identify salient student characteristics, such as gender and socioeconomic status, which likely moderate the transactional relationships. The examination of gender differences is warranted in the timing and nature of progressions, as prior longitudinal studies have indicated gender-specific transactional relationships (e.g., [19,26,30]) in different developmental stages. Practitioners may benefit from learning about whether these are gender-specific pathways between behavior and academic performance, particularly regarding internalizing symptoms in girls. Further research can look into how varying socioeconomic status affects the association between behavior and academic performance. In addition, racial/ethnic differences are likely to be at play, given the deeply rooted structural racism in social institutions in the US, such as the education system [49].

Future research can also identify mediators in the bidirectional models to unpack the developmental dynamics in the early school years. Developmental variables, such as interpersonal social skills and peer competence, have been strongly implicated in the association between symptoms and adaptive behavior [5,50]. In addition, Metsäpelto et al. [21] also indicated the mediating role of task-avoidant behavior on the pathways between externalizing problems and academic performance.

We limited the scope of work and focused on two domains of problem behaviors and one major academic performance measure (composite of reading and math). There was a notable degree of difference between the conceptualization of behavioral constructs. Different types of internalizing symptoms may yield different associations with academic outcomes. For instance, while depression and anxiety both fall under the umbrella term “internalizing disorders,” studies that examined these in isolation were more likely to produce findings that diverged from the trends identified in studies that examined internalizing symptoms as a singular construct. Future research can examine the relationships between specific internalizing behavior, such as depression and anxiety, and specific domains of academic performance (reading and math). In addition, children’s problem behaviors were solely based on the ratings reported by teachers, which might introduce assessment bias. Future research can collect multiple sources of data to triangulate the findings on children’s problem behaviors.

## 5. Conclusions

This longitudinal study indicated bidirectional relationships between behavioral and academic problems, with externalizing and internalizing problem behavior predicting academic performance and vice versa in a nationally representative student sample in early childhood. The findings of the current study help enhance the understanding of how academic and behavior challenges co-interact over time and suggest that, early on, teachers can identify and support students who experience academic and behavioral difficulties. 

## Figures and Tables

**Figure 1 ijerph-19-09583-f001:**
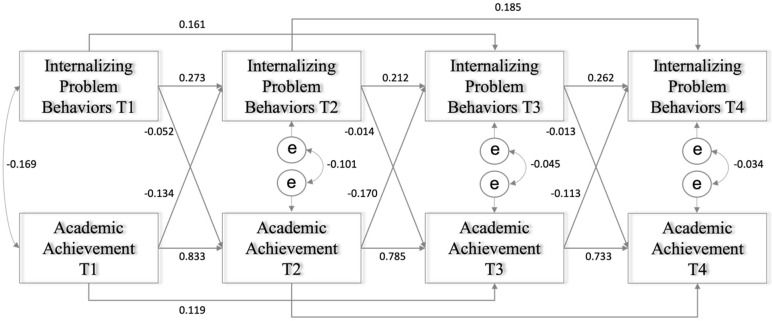
Transactions between internalizing problem behaviors and academic achievement.

**Figure 2 ijerph-19-09583-f002:**
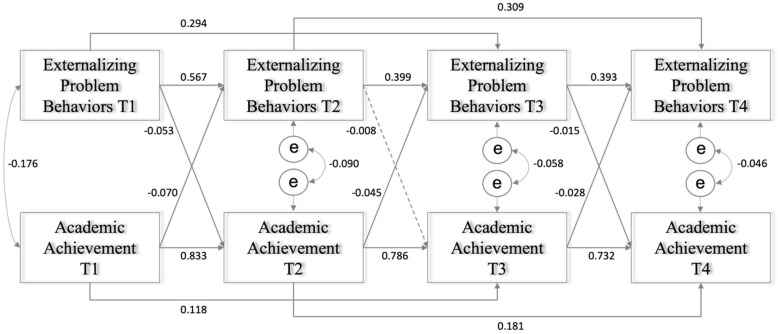
Transactions between externalizing problem behaviors and academic achievement (dotted lines indicate nonsignificant paths).

**Table 1 ijerph-19-09583-t001:** Descriptive statistics of problem behaviors and academic performance variables.

Variable	*M*	*SD*	Skewness	Kurtosis	Min	Max
Internalizing problem behaviors T1	1.50	0.49	1.36	2.49	1	4
Internalizing problem behaviors T2	1.54	0.50	1.41	2.78	1	4
Internalizing problem behaviors T3	1.59	0.53	1.28	2.10	1	4
Internalizing problem behaviors T4	1.60	0.54	1.29	1.97	1	4
Externalizing problem behaviors T1	1.62	0.62	1.21	1.40	1	4
Externalizing problem behaviors T2	1.72	0.61	1.10	1.06	1	4
Externalizing problem behaviors T3	1.72	0.62	1.07	0.81	1	4
Externalizing problem behaviors T4	1.69	0.62	1.14	1.06	1	4
Academic achievement T1	59.73	12.68	0.65	1.22	22.14	122.71
Academic achievement T2	83.83	15.29	−0.19	0.15	22.50	136.68
Academic achievement T3	101.13	15.92	−0.39	0.05	32.95	142.37
Academic achievement T4	112.31	15.27	−0.47	−0.22	44.22	152.13

Note. T1: kindergarten; T2: 1st grade; T3: 2nd grade; T4: 3rd grade.

**Table 2 ijerph-19-09583-t002:** Bivariate correlations.

	1	2	3	4	5	6	7	8	9	10	11	12
1. Internalizing problem behaviors T1	—											
2. Internalizing problem behaviors T2	0.30	—										
3. Internalizing problem behaviors T3	0.26	0.30	—									
4. Internalizing problem behaviors T4	0.26	0.29	0.35	—								
5. Externalizing problem behaviors T1	0.27	0.18	0.19	0.19	—							
6. Externalizing problem behaviors T2	0.11	0.30	0.20	0.20	0.60	—						
7. Externalizing problem behaviors T3	0.11	0.14	0.33	0.19	0.56	0.60	—					
8. Externalizing problem behaviors T4	0.08	0.14	0.17	0.31	0.53	0.58	0.61	—				
9. Academic achievement T1	−0.17	−0.17	−0.20	−0.19	−0.18	−0.17	−0.16	−0.14	—			
10. Academic achievement T2	−0.19	−0.21	−0.23	−0.21	−0.20	−0.21	−0.19	−0.17	0.85	—		
11. Academic achievement T3	−0.19	−0.20	−0.23	−0.20	−0.19	−0.19	−0.19	−0.17	0.79	0.90	—	
12. Academic achievement T4	−0.18	−0.20	−0.22	−0.20	−0.18	−0.19	−0.19	−0.18	0.75	0.85	0.91	—

Note. All the correlation coefficients are statistically significant at *p* < 0.001. Transactional relationships between internalizing problem behaviors and academic performance.

## Data Availability

Not applicable.

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
