# Peer review of "Transactions between Problem Behaviors and Academic Performance in Early Childhood"

_ijerph, 2022, doi:10.3390/ijerph19159583_

Round 1
Reviewer 1 Report
This article fills the gap in the research, first of all contributing to the understanding of transactional relationships between behaviors and academic performance in early childhood. The literature referenced and the model for the research used are well selected. There are a few suggestions for authors to make minor improvements to the article:
1. Sample - Why do the authors state an approximate number of respondents and not an exact number?
2. Results - Both Figure 1 and Figure 2 deserve being presented in a higher resolution to improve readability.
3. Discussion - The results of this research are quite consistent with the authors' assumptions. Yet „externalizing problem behaviors significantly predicted academic achievement at time 1 and time 3 but not at time 2“ was found. What is the author's explanation for this finding?
Reviewer 2 Report
This manuscript is very informative aimed to analyse and describe the complex associations between behavioral problems and academic achievement in early childhood. I would highlight a couple of weaknesses in my comments and suggestions, needing minor revision in order to improve the quality of the manuscript.
Introduction
I would kindly suggest to the authors to state earlier in the text in the Introduction what would be internalizing and externalizing problem behaviors as they stated in the abstract.
The Introduction is too long and not easy to read. I would suggest starting the paper from “Current study” paragraph. The authors could add another paragraph about internalizing and externalizing problem behaviors – having whole section with several paragraphs only distracts the reader from the scope of the survey.
Method
There is necessity to state the exact number of the subjects in the survey. Stating that the sample involves approximately 18.000 children is not acceptable for the method of scientific journal.
Table 1 is not self-explanatory and clear – the authors should explain what “Unweighted” and “Weighted” means. Also, I suppose that first column represents number (n), and second frequency (%). Moreover, the percentages from the Table 1 are repeating in the text, but the text should only add to the table. I suggest to the authors to decide whether to keep the table and describe the sample roughly it in the text. Or if the authors decide to exclude the table, then the text should have both number of children and percentages (I suggest having in brackets).
The Methods needs a paragraph on hoe the representative sample was achieved, inclusion, exclusion criteria, how the children were enrolled.
Since the authors stated in the Method that the teachers rated children behavior, then it should be explained how many teachers, and how they were enrolled, and what were the exclusion and inclusion criteria. Also state if the teachers were calibrated.
Results
Table 2 is not clear; please consider making it simpler for the reader. Also if the authors decided to analyze children according to the age groups T1, T2, T3 and T4, please explain and state that in the method section.
Overall statement
The topic of the manuscript is relevant, and I recommend publication with major revision.
Round 2
Reviewer 2 Report
This manuscript is very informative aimed to analyze and describe the complex associations between behavioral problems and academic achievement in early childhood. Although I appreciate all the changes and efforts authors performed during review in order to improve quality of the manuscript, I would highlight following important methodological weaknesses that need to be addressed in order to have methodological soundness of the manuscript.
Round 1, Method, Comment 2: There is necessity to state the exact number of the subjects in the survey. Stating that the sample involves approximately 18.000 children is not acceptable for the method of scientific journal.
Thank you for the recommendation to include an exact sample size. According to the reporting policy of the ECLS-K dataset, we are required to rounded all of the numbers to the nearest 10. Therefore, we revised and included unweighted sample size rounded to the nearest 10.
“This study consisted of approximately 18,140 children, derived from a nationally representative sample in the United States participating in the Early Childhood Longitudinal Study, Kindergarten Class of 2011 (ECLS-K: 2011).” (page 5, line 208)
Round 2: There is absolute necessity to state the exact number of subjects included in the sample size, not “approximately”. I would kindly ask authors to rephrase this in the method section and to add the exact value. Also, please add the average age of the subjects.
Round 1, Method, Comment 5: Since the authors stated in the Method that the teachers rated children behavior, then it should be explained how many teachers, and how they were enrolled, and what were the exclusion and inclusion criteria. Also state if the teachers were calibrated.
Thank you for pointing out the need for clarification. Based on the design of the ECLS-K study, data from the study are not representative of teachers. They are included in the study by virtue of their connection to the sampled children. The decision to include or exclude a teacher to be in the study lies in whether the school with which a teacher is affiliated is selected and agrees/disagrees to participate in ECLS-K and also whether a child (his/her parents) agrees to patriciate in ECLS-K or not. Teachers who completed the survey just happened to be the instructor of the child who was in the study with the permission from his/her parents. Therefore, there is no specific procedures implemented to recruit teachers.
Round 2: Having in mind that teachers rated the children behavior they cannot be considered not representative. I would kindly ask authors to add the explanation they provided above in the method section, so this could be clear to the readers. Also, if the teachers performed “child assessments of reading and mathematics and teacher questionnaires of children’s internalizing and externalizing problem behaviors.” as stated in the text, I was wondering how the authors excluded teachers’ bias, and if not, this should be discussed under limitations of the study in the discussion section.
